# Relationship between the Effects of Perceived Damage Caused by Harmful Rumors about Fukushima after the Nuclear Accident and Information Sources and Media

**DOI:** 10.3390/ijerph20032077

**Published:** 2023-01-23

**Authors:** Chihiro Nakayama, Hajime Iwasa, Nobuaki Moriyama, Seiji Yasumura

**Affiliations:** Department of Public Health, Fukushima Medical University School of Medicine, Fukushima 960-1295, Japan

**Keywords:** Fukushima nuclear accident, information, mass media, Internet, harmful rumor

## Abstract

The nuclear accident that accompanied the Great East Japan Earthquake of 11 March, 2011, was also an information disaster. A serious problem that arose after the accident and persisted for a long time was the damage caused by harmful rumors (DCBHR). In 2016, a cross-sectional questionnaire survey on health and information was conducted in Fukushima. The eligible population of this survey was 2000 Fukushima residents, which included those in the evacuated areas. We received 861 responses. Data were analyzed using the responses to the question about perceived DCBHR as the objective variable and the sources of information residents trusted and the media they used as explanatory variables. A multiple logistic regression analysis revealed that those who trusted government ministries and local commercial TV were significantly associated with no effect. In contrast, those who used Internet sites and blogs were significantly associated with a negative effect. This study underlines the pivotal importance of media and information, literacy, and education and discusses how these should be improved to avoid DCBHR in the future. Furthermore, accurate information should be made available to all sections of the population to diminish DCBHR.

## 1. Introduction

The Tokyo Electric Power Company’s Fukushima Daiichi Nuclear Power Plant accident (hereinafter referred to as the “Fukushima nuclear accident”) caused by the Great East Japan Earthquake on 11 March, 2011, resulted in the spread of radioactive materials throughout the region. Residents suffered both extensive physical and psychological damage after the accident. Furthermore, a significant social problem that occurred after the accident and continued for a long time was the damage caused by harmful rumors (DCBHR), called “fuhyohigai” in Japanese [1,2]. Fuhyohigai has a distinct meaning [1,3]; it is sometimes translated as “harmful rumor” or “reputational damage” in English; it combines “fuhyo”, which means “harmful rumor” and “higai”, which means “damage”.

In August 2011, the Governor of Fukushima Prefecture issued the following message. “On 11 March 2011, Fukushima Prefecture was hit by a massive earthquake and tsunami that triggered the accident at Tokyo Electric Power Company’s Fukushima Daiichi Nuclear Power Station and subsequent damage to the prefecture’s reputation due to harmful rumors” [4]. The term DCBHR has been used in the past in relation to nuclear accidents to describe the economic damage suffered despite the safety of food, commodities, and land, among others [3]. The word fuhyo was first used in Japan in 1956 in parliament. At that time, the price of tuna and other seafood decreased in connection with the US hydrogen bomb test in the Marshall Islands in 1954. Various compensations were given to fishermen by the government to minimize what was referred to as “indirect damage”. In 1974, there was a radioactivity leak from the nuclear-powered ship Mutsu. Although nearby seafood was not affected, once more, prices dropped. At the same time, the “Fish Price Stabilization Fund” was distributed to Aomori Prefecture, where the accident occurred, with the aim of stabilizing fish prices based on harmful rumors related to ‘Mutsu’. In the late 1990s, the term “DCBHR” started being used. In 1997, when a cargo ship ran aground in the Sea of Japan causing a heavy oil spill, the news report used the term “heavy oil spill in the Sea of Japan”, even though only part of the area was contaminated. As a result, seafood from the Sea of Japan, which was not polluted, fell in price and resellers refused to buy it. At this time, the term DCBHR was widely used in the mass media (newspapers, television, and magazines), and since then, it has taken root in Japanese society. It has been used in other contexts, such as disasters and infectious diseases [1,3,5]. In February 1999, the news sensationally reported dioxin contamination of vegetables in Tokorozawa City, and the price of vegetables produced in the city plummeted. The affected farmers sued the TV station for defamation and economic damage. In 2004, the supreme court ruled in favor of the plaintiff, and the TV station apologized to the farmers and paid settlement money [6].

In July 2011, the Ministry of Education, Culture, Sports, Science and Technology (MEXT) of Japan listed the avoidance of food, agricultural and marine products, goods, and land, including by the residents of Fukushima Prefecture and surrounding areas, as an example of so-called DCBHR related to the nuclear accident in Fukushima Prefecture [7]. In August 2011, the Fukushima Prefectural Government stated that, as a result of DCBHR, all industries in Fukushima, including the key industries of agriculture, forestry, fishery, manufacturing, commerce, and tourism, were on the verge of extinction due to reputational damage. The following are some examples of the damage: (1) reports of child bullying at evacuation centers caused by the belief that radiation could be transmitted; (2) evacuees from Fukushima Prefecture not being accepted for relocation; and (3) signs at gas stations that read “Fukushima Prefecture residents not welcome” and cases where vehicles and trucks with Fukushima license plates were refused service at stores outside Fukushima Prefecture. Additionally, as an emergency message concerning harmful rumors, Fukushima Prefecture announced that baseless assumptions and discrimination against radiation victims were a violation of human rights [4]. Ohto et al. noted that malicious rumors (fuhyo, in Japanese) about Fukushima, such as the land being too ruined to inhabit, inevitable death by radiation, the birth of deformed babies, and foods being too dangerous to eat, spread across Japan and the world [8]. In Fukushima, DCBHR is not only plaguing producers but also distributors, sellers, travel agents, consumers, and even the general population [8].

Taking the aforementioned cases as a whole, the definition of DCBHR derived from them designates the economic damage caused by people or organizations that, owing to the extensive media coverage of certain social problems (incidents, accidents, environmental pollution, disasters, and recessions), regard things (food, products, land, and companies) that were originally considered “safe”, or whose risks are almost negligible, as dangerous and stop consuming or trading them. This term also includes psychological damage [1,3,5]. As the word “safe” means to think of oneself as safe or to be judged safe by someone, the term DCBHR involves subjective judgments, and there is no consensus regarding its meaning. In some respects, the perception of the victim determines the existence and nature of DCBHR. Therefore, the meaning of the term may change depending on the user [1]. In addition, DCBHR is almost always caused by press reports, not rumors. Furthermore, it is not necessarily caused by misinformation in news reports. For example, although a report on the “Sea of Japan Heavy Oil Spill Accident” from 1997 was not completely false, sensationalized language increased anxiety among readers, thereby causing DCBHR. This resembles the concept of “social amplification of risk” presented by Kasperson et al. [2,3,5,9].

Since the 2016 US presidential election and Brexit in the UK, disinformation, which is often referred to as “fake news”, has spread, particularly on social networking sites (SNS) and the Internet, becoming a serious problem [10]. The definition of “fake news” is fabricated information that mimics news media content in form but not in organizational process or intent. Fake news overlaps with other information disorders, such as misinformation (false or misleading information) and disinformation (false information that is purposely spread to deceive people) [11]. In Japan, it can be said that a precursor to this fake news phenomenon was seen in the DCBHR regarding the 2011 Fukushima nuclear accident. For example, in 2013, a map depicting the height of the tsunami in the Pacific Ocean was spread around the world via the Internet, supposedly showing the flow of the radioactive material released [12,13]. Originally from a news account in the US, it was widely viewed in Japan as well. This instance of DCBHR indirectly caused economic and psychological damage by making Fukushima look dangerous. The Fukushima nuclear accident was the first nuclear accident in which the Internet was widely used, and it is likely that the media coverage that could lead to DCBHR included online information as well as traditional mass media such as newspapers and television.

Aside from the issues of discrimination and human rights violations, there is some uncertainty as to whether DCBHR refers to perceived damage or actual damage. For example, it was not known whether agricultural and marine products from Fukushima were safe unless their level of radioactivity was measured. For this reason, as previously mentioned, in July 2011, the Ministry of Education, Culture, Sports, Science and Technology, referred to so-called DCBHR [3]. Immediately after the nuclear power plant accident, the Japanese government set a provisional limit of 500 becquerels per kilogram of radioactive material in food. Then, about a year after the accident, the government set a standard level of 100 becquerels per kilogram, which is considered rigorous, even by global standards. Subsequently, the possibility of hazardous foods being released into the market was almost eliminated [14,15]. Furthermore, the inspection system was further improved, and in 2013, most of the agricultural products from Fukushima were categorized as “radiation not detected” (ND). Additionally, in 2012, the Fukushima Prefecture started to inspect all bags of rice harvested throughout the region, and between 2014 and 2019, none was found to have excessive levels of radiation. This indicates that food items from Fukushima were considered “safe” for consumption. For these reasons, we believe that consumers understood what was factual and what was DCBHR in 2016 when this survey was conducted.

However, DCBHR continued, and even in 2021, shipments of agricultural and marine products from Fukushima had still not recovered to pre-disaster levels. The gap between the price of products from Fukushima and the national average is gradually narrowing, but some commodities from Fukushima, such as beef and peaches, are still below the national average [16]. A survey conducted in Tokyo in June 2021 found that 20.7% of respondents were hesitant to eat food items produced in Fukushima Prefecture owing to concerns regarding radiation [17]. The problem with the current DCBHR is the economic damage that occurs when almost all food products in circulation are ND. Furthermore, the psychological damage is significant and contributes to the stigma faced by Fukushima residents [18,19]. For example, more than 30% of the residents of Fukushima are concerned about the harmful impact of low-dose radiation exposure on their children and future grandchildren, despite the fact that no harmful impact has been found. Prior studies have shown that this is due to the influence of TV reports and other sources of information [20].

Considering all the above, the term DCBHR, which is unique to Japan, has a long history, and although its definition has gradually changed, it is now firmly established in Japanese society, with government policies, such as compensation and other measures, having been implemented to support victims of it. One of the causes of DCBHR is media coverage, and some victims have filed lawsuits against TV stations and won their cases. It can be said that, to some extent, victims’ perceptions determine the existence and content of DCBHR. However, no research has yet been conducted on the causes of DCBHR regarding the Fukushima nuclear accident among the residents of Fukushima. Therefore, this study set out to examine this issue. A prior study conducted a survey on health and information in 2016 involving 2000 Fukushima residents (aged between 20 and 79 years) to determine the relationship between Fukushima residents’ anxiety about radiation, the sources of information about radiation they trusted, and the type of media they used. The questionnaire used comprised 35 questions, one of which was “Has DCBHR affected your life after the nuclear accident?” [21]. We hypothesized that the answer to this question would determine the presence or absence of perceived DCBHR.

This study proposed to answer the following research question: What were the causes of the DCBHR perceived by Fukushima residents regarding the 2011 Fukushima nuclear accident?

Further, we hypothesized that the presence or absence of a perceived impact of DCBHR on Fukushima residents would be related to the media they use and the sources of information they trust, particularly online information. Consequently, this study aimed to explore the relationship between the effect of perceived DCBHR as the objective variable and the sources of information on radiation that the residents trusted and the media that they used as explanatory variables. Our ultimate goal was to determine ways to reduce this damage.

## 2. Materials and Methods

This study was based on a questionnaire survey of residents in Fukushima Prefecture. The study methods were approved by the Fukushima Medical University’s Ethics Committee (Approval number: 2699). We considered a returned questionnaire as an indication of consent regarding the study’s objectives and to voluntarily participate in it.

### 2.1. Participants

This survey was conducted with 2000 residents of Fukushima Prefecture aged between 20 and 79 years. We divided Fukushima Prefecture into four areas based on the general regional classification, i.e., Aizu, Nakadōri, Hamadōri, and the evacuation area (the restricted area, evacuation prepared area, and deliberate evacuation area, as determined on 22 April 2011), and selected 500 people from each area (Figure 1). The selection was based on two-stage stratified random sampling (stage one: a survey of the region; stage two: a survey of individuals randomly selected from the Basic Resident Registration). In the first stage, the four regions were first divided into three strata: cities with a population of 100,000 or more, cities with a population of less than 100,000, and towns/villages. For each stratum, a sample size was allotted based on the proportion of the total population. Next, the number of survey spots was determined for each stratum in the four areas, with a sample size of 33 or 34 per survey spot. Taking the districts of the 2010 National Census as survey spots, allotments were made to each stratum using random sampling. In the second stage, 33 or 34 people were selected from each survey spot using the Basic Resident Registry. Nakadōri and Hamadōri included local municipalities that were partially in the evacuation area; as such, these were included in the evacuation area. The instrument used, entitled “Survey of Health and Information”, was administered as an anonymous, self-reported postal questionnaire. We received 916 responses from 1985 survey participants (excluding those that were returned to sender in cases whereby no one resided at the address). After the 55 respondents who left the fields for age or gender were excluded, we analyzed data from 861 respondents, with a valid response rate of 43.4%.

### 2.2. Survey Variables

The answer to the question “Has the damage caused by harmful rumors affected your life?” was considered the objective variable. The respondents selected from the following options: “no”, “somewhat”, or “yes”. Those who chose the last two options were asked to freely describe the content.

Answers to several questions that focused on sources of information to trust were considered explanatory variables. Participants were asked to select up to three main items from the following 11 options: International organizations (United Nations (UN), World Health Organization (WHO), etc.), experts from universities and other academic institutions, government ministries, local newspapers, national newspapers, NHK (public broadcasting), private local broadcast television, private national broadcast television, local government, volunteer organizations such as citizens’ groups, and “none of the above”. Participants were asked to indicate the type of media used to obtain information on radiation by selecting up to three main items from the following 13 options: local newspapers, national newspapers, NHK television (public broadcasting), private local broadcast television, private national broadcast television, radio, Internet news, Internet sites and blogs, SNS (Facebook, Twitter, etc.), magazines/books, local government publications, word of mouth, and “none of the above”. These sources of information that people trusted and the media they used were utilized as explanatory variables.

The following were considered as other variables. For demographic information, respondents were asked about their age, gender and area of residence, and their resident status before and after the accident. They chose from the following options: own home, public housing, government-subsidized housing, rental home or apartment, temporary housing, and home of a friend/relative or other. Respondents were asked to specify whether they had children aged 18 or younger, 19 or above, were pregnant, had a pregnant family member, or “none” (no children) at the time of the earthquake. They were also asked about their educational background, employment status, family structure, housing prior to the earthquake, and whether they had relocated to avoid radiation.

Questions also included indications of social capital and participation in local groups as well as health-related behaviors, such as exercise, sleep satisfaction, alcohol consumption, and smoking. The respondents were asked to rate their health status. A prior study used the health literacy scale to measure communicative and critical health literacy, developed by Ishikawa et al. to be used by the general public [22]. The Cronbach’s alpha coefficient of the scale was 0.86 in Ishiwaka et al.’s study [22] and 0.88 in the present study. Knowledge of radiation was assessed according to the respondents’ knowledge of the following five areas: awareness of the properties of radiation, probability of death from cancer, genetic impact, DNA repair, and food reference values. Participants were asked to rate short sentences, which were well-known and often misunderstood, as being either true or false.

A specific group of health-related questions focused on anxiety. Participants were asked to rate their level of health-related anxiety immediately after the nuclear accident and the current level related to the effects of radiation on their health due to the nuclear accident. Questions were asked about receiving health examinations or attending lectures on radiation (eight items). Participants were asked to rate their radiation anxiety using the seven-item Radiation Anxiety Scale developed by Umeda et al. [23]. Participants were asked about their choice of actions to protect themselves against radiation (four items). They were also asked to indicate something gained through the earthquake experience, such as post-traumatic growth, which can be seen as a positive psychological change experienced as a result of a struggle with a major life crisis or a traumatic event.

### 2.3. Analysis Plan

To explore the correlation between “the presence of the effect of DCBHR in your life” and other items, we first performed a univariate analysis (Student’s *t*-test, and Chi-squared test), with “the presence of the effect of DCBHR in your life” as the objective variable and all other responses as explanatory variables. Statistical significance was evaluated using two-sided, design-based tests with a 5% level of significance.

Many survey answers were merged into two or three categories for analysis. For example, participants were asked to rate the effects of DCBHR on a scale comprising “none”, “a little”, and “yes”. These responses were treated as two variables: “a little or lot” and “none”. The “presence or absence of effects of the reputational damage in your life” was used as the objective variable.

For age, respondents were divided into three groups: 20–44 years (young adults), 45–64 years (middle aged), and 65 years and older (older adults). For area, two groups were created, comprising those located in the evacuation area or “other” (Aizu, Nakadōri, and Hamadōri). Respondents specified whether they had children, and, for the analysis, they were divided into those with and without children.

For health literacy, scores were determined as the sum of the values of the five items, with those above the mean placed in the “high” group and those at the mean or below placed in the “low” group.

Thereafter, a multivariable logistic regression analysis was performed using the explanatory variables found to be significant in the univariate analysis with “the presence of the effect of DCBHR in your life” as the objective variable.

Questions regarding trusted sources of information and media used to obtain information on radiation elicited multiple answers (participants were instructed to choose three main items). Owing to the restriction on the number of items that could be selected, these explanatory variables were not completely independent of each other. However, this had an insignificant impact. Rather, we thought it was important to see the confounding factor between “trusted sources of information” and “media used”. Therefore, we put the variables that were significant in the univariate analysis into a single multiple logistic regression model to examine their association with the presence of the effect of the perceived DCBHR in participants’ lives.

Finally, information gained from government ministries, trusted private local broadcast television, national newspapers, and relevant sites and blogs were used as explanatory variables. Young adults, evacuation area, no children, above high school graduation, member of the neighborhood association and/or civic organization and vocational organizations, and knowledge of food reference values were used as moderator variables. Not being a member of an organization, which was highly correlated with being a member of the neighborhood association, was excluded (correlation coefficient: −0.73). We also excluded relocation to avoid radioactivity, which was highly correlated with the evacuation area (correlation coefficient: 0.69). Moreover, gender was added as a basic moderator variable.

## 3. Results

There were 861 valid responses, representing a valid response rate of 43.4%. In total, 12 people did not respond to the question on perceived DCBHR, which resulted in 849 total responses for analysis.

The mean age (years ± SD) of the respondents was 56.4 ± 14.8 years. Basic information about the respondents is presented in Table 1.

Table 2 shows the distribution of responses to the question “Has DCBHR affected your life after the nuclear accident?” by region. The proportion of those who answered “yes” was 71.7% in the evacuated areas, which was significantly higher than that in the other areas (*p* < 0.001).

Participants’ trusted sources of information and media sources used are summarized in Figure 2. Participants were instructed to choose three main items, which made the total exceed 100%.

Table 3 and Table 4 show the results of the univariate analysis (chi-squared test) conducted to examine the relationship between “the effect of DCBHR” and all sources and media used to receive information on radiation. The proportion of those affected by perceived DCBHR was significantly lower among those who had trust in government ministries, private local broadcast television, and national newspapers. In contrast, the proportion was significantly higher among those who used Internet sites and blogs.

Table 5 shows the results of the univariate analysis (chi-squared test) conducted to examine the relationship between “the effect of perceived DCBHR” and all other items. This table only includes items that were significant.

The proportion of those affected by perceived DCBHR was significantly lower among those who belonged to the Young Adult group (20–44 years old), did not have children at the time of the disaster, and were educated at junior colleges and vocational schools or above. In contrast, it was significantly higher among those who were in an evacuation area, participated in neighborhood associations and/or voluntary organizations, such as NPOs and civic activities, and vocational organizations, and answered the question on the standard value of food correctly.

Table 6 shows the results of the multiple logistic regression analysis with the presence or absence of the effect of perceived DCBHR as the objective variable.

Those who trusted government ministries (odds ratio: 0.67; 95% CI: 0.48–0.93) and/or private local broadcast television (0.63; 0.42–0.93), used Internet sites/blogs (2.50; 1.20–5.20), were in the evacuation area (2.17; 1.48–3.20), did not have children at the time of the earthquake (0.62; 0.46–0.84), were educated at junior colleges and vocational schools or above (0.70; 0.52–0.94), were a member of a vocational association (2.46; 1.33–4.52), and answered the question on the reference value of food correctly (1.43; 1.10–1.93) were significantly associated with the “presence of the effect of DCBHR”.

## 4. Discussion

According to the demographics of the Fukushima Prefectural Population Survey Annual Report 2015 [24], it was estimated that individuals in the Fukushima Prefecture were aged 51.6 years on average in 2016. The proportion of males and females was 49.5% and 50.5%, respectively. The sample in the current study was largely representative of the established demographics for age and gender.

The proportion of those who answered “yes” to the question on the effect of perceived DCBHR was significantly higher in the evacuation area than in other areas. In the evacuation area, the radiation dose due to the nuclear accident was relatively higher. Furthermore, since an evacuation was actually required, it was probable that the effect of perceived DCBHR was significant.

The hypothesis that the presence or absence of a perceived impact of DCBHR on Fukushima residents would be related to the media they use and the sources of information they trust, particularly online information, was supported. The results of the multiple logistic regression analysis (Table 6) showed that those who trusted government ministries were significantly associated with the absence of the effect of perceived DCBHR (OR: 0.67). This may be because the government ministries collected extensive data and worked to dispel harmful rumors about agricultural and fishery products and tourism. In a previous study that used data from the same Health and Information Survey, it was found that those who trusted government ministries had lower anxiety regarding the effects of radiation on their health [21]. Our results were also partially consistent with this finding.

Similarly, those who trusted the information disseminated by private local broadcast television were significantly associated with the absence of the effect of perceived DCBHR (OR: 0.63). Since private local television stations reported the facts with detailed local data, it was thought that those who trusted in it were less likely to think that they were affected by perceived DCBHR. This finding was partially in accordance with that of a prior study that showed that those who used private local broadcast television had lower anxiety regarding the effects of radiation on their health [21].

Contrastingly, the response from those using Internet sites and blogs was the only variable among the “information source/media” that was significantly associated with the presence of the effect of perceived DCBHR (OR: 2.50). Furthermore, this variable had the highest OR of all the factors. Previous studies reported that the Fukushima nuclear accident created distrust toward mass media [1]. People who were distrustful of mass media used the Internet to obtain information on the accident that could not be conveyed on television or in newspapers [25]. A previous study reported that some mothers who were anxious regarding food items produced in Fukushima and felt that the public information was inaccurate and the information on the Internet was correct, voluntarily evacuated from the non-evacuation areas in Fukushima [26]. Furthermore, those who used Internet sites and blogs had higher anxiety regarding the negative effects of radiation on their health [20]. Another previous study conducted in Spain also showed that the frequency of Internet use is significantly associated with health worries, including worry about radiation [27].

In a nuclear disaster, information overflows, not only from the mass media, but also from other sources, which makes it difficult to understand what is correct [28]. On the Internet, an anonymous individuals and organizations can disseminate information in an almost unlimited manner, without scrutiny or editorial moderation, and spread it rapidly [29,30]. Therefore, it is probable that after the Fukushima nuclear accident, disinformation and rumors were disseminated on the Internet. This was thought to be the reason why those who used Internet sites and blogs had a significantly higher OR.

Those who did not have children at the time of the disaster had heard fewer harmful rumors related to children compared to those who had children, and as such, their OR was significantly lower (OR: 0.62). Furthermore, a previous study reported that many people believed that low-dose radiation had a greater impact on children than on adults [31]. Thus, people with children often tried to avoid agricultural products produced in the Fukushima Prefecture. Moreover, another study showed that the factor “living with children” was associated with people refraining from purchasing food items produced in affected areas to avoid radioactive materials [32].

The OR for those educated at junior colleges and vocational schools or above was significantly lower (OR: 0.70). The reason for this was thought to be that the proportion of people not affected by harmful rumors was higher owing to the acquisition of scientific knowledge and literacy through higher education [33].

Those in the evacuation area received plenty of negative information and rumors regarding the doses of air radiation, food items contaminated with radioactive substances, and health hazards caused by exposure in the area where they lived. Thus, their OR was significantly higher (OR: 2.17). It was also possible that there was prejudice or discrimination against evacuees following their relocation to other prefectures.

The OR of those who participated in vocational organizations, such as business associations, peer associations, industry groups, labor unions, and so on, was significantly higher (OR: 2.46). DCBHR, by its original definition, implies that a product cannot be sold, or its price is reduced, owing to false rumors. Therefore, it has a great impact on people involved in businesses. Many food producers, especially those that participated in shops and trade associations, may have had the experience of not being able to sell their produce. They may also have been concerned about the harmful rumors that were having a direct negative impact on the sales and prices of their product. As such, their OR was significantly higher [2].

There are several possible reasons for the significant relationship between having knowledge of “food reference values of radioactive materials” and the presence or absence of “the effects of perceived DCBHR” (OR: 1.43). Those who avoided food items from Fukushima may have been extremely interested in “food safety” and may have had knowledge of “food reference values of radioactive materials”. However, they may still have doubted the reference values or the inspection system itself. Alternatively, even if the food items contained radioactive substances below the reference value, such people considered that food items produced in Fukushima should be avoided owing to the fear that radioactive substances might enter the body, albeit in very low quantities. Previous studies have shown that some people refuse to accept materials even when their radioactivity is below the reference value [32,34,35].

It is probable that the producers and sellers of food items who were affected by perceived DCBHR were familiar with the “food reference values of radioactive materials”. Furthermore, it is also probable that ordinary people, i.e., not sellers or producers but individuals who knew the “food reference values of radioactive materials”, were conscious of how producers and sellers of foods were being affected. Nevertheless, no food items on the market exceeded the reference values. As such, the effects of DCBHR were observed; in other words, not even safe products could be sold [2]. It is also possible that those who had been affected by DCBHR learned of the food standards afterward.

Regarding the question “Has DCBHR affected your life since the nuclear accident?”, 59.5% of those who answered “a little” or “yes” provided context in the form of additional comments. Although this was not a breakdown of all the answers, it was useful as a source of further information for consideration.

“I avoided food items from Fukushima because I was concerned about radiation” was the most common response. This behavior fits the orthodox definition by researchers regarding the behavior of consumers [2]. Although we cannot ascertain whether this behavior was in the past or reflective of participants’ current attitude when they answered the questions, the proportion of those who adopted this behavior decreased. A total of 36.1% of the study respondents reported having avoided foods from Fukushima in the past but no longer doing so at the time at which this survey was conducted. However, 30.6% continued to avoid foods from Fukushima at the time of answering this survey.

The second most common item related to DCBHR, i.e., “Consumers did not buy food items from Fukushima”, also fits the definition. In this case, harmful rumors brought about behavior among consumers that resulted in economic damage to food producers and sellers. The respondents included some food producers and sellers who described their personal experiences.

The third most common response, “I have experience of prejudice and discrimination”, was different from the orthodox definition. In other words, those who were influenced by the harmful rumors caused damage and distress to respondents through discrimination and prejudice. The fourth most common item, “damage to work and economy”, fits the definition. The fifth most common item, “others”, included a wide range of topics, such as a decrease in the number of tourists, moving and evacuating, family separation, divorce, suicide of acquaintances, and so on.

The respondents in this survey had a broader view of the types of DCBHR compared to its standard definition. It was evident that DCBHR in relation to the Fukushima nuclear accident gave rise to human rights violations, such as prejudice and discrimination against the locals of the prefecture. This observation was consistent with the findings presented in a previous study [31].

### 4.1. Implications

This study showed that the presence or absence of the effects of perceived DCBHR differed based on the sources of information trusted by respondents and the media consumed to obtain information about radiation. Above all, it was suggested that misinformation, which was rampant on the Internet, had a great impact on perceived DCBHR. Contrastingly, private local broadcast television provided accurate information and played a role in controlling perceived DCBHR. This means that the integrated use of traditional and social media is necessary as means of effective communication in nuclear and radiological emergencies [36].

Furthermore, it is necessary to improve media literacy so that accurate and factual information can be identified. This requires media literacy education. In the future, in major disasters such as nuclear accidents, there should be room to consider legislation to discourage the propagation of misinformation [28].

Information from government ministries was shown to be important to reduce perceived DCBHR; however, only 25% of the respondents trusted such information. Previous studies also reported that the credibility of government ministries and agencies was low in Fukushima [31]. Government ministries and agencies must strive to increase their credibility by increasing transparency.

The OR for those who were educated beyond a high school level was significantly higher. It was suggested that making science education compulsory may shield against DCBHR. In today’s world, the amount of information that requires scientific knowledge to understand it is steadily increasing. Therefore, the enhancement of education related to science, especially radiation, is desirable.

### 4.2. Limitations

Our study has a few limitations. First, since this was a cross-sectional study, causality could not be determined. Second, since respondents tended to be relatively older, our study population included fewer users of SNS, which were said to be associated strongly with disinformation and harmful rumors. Third, the response rate was 43%, and it was not certain whether the results reflected the overall situation. Particularly, those with poor physical or mental health were generally less likely to respond to the survey, which may have influenced the overall results.

Despite these limitations, we were able to demonstrate the association between the effect of perceived DCBHR and the sources of information and media consumed regarding radiation. This is the first study to explore such a relationship in relation to the Fukushima nuclear disaster in 2011.

## 5. Conclusions

In this study, we examined the relationship between the effect of perceived DCBHR and information sources and media. Notably, those who trusted government ministries and those who trusted private local broadcast television were significantly associated with the absence of such an effect. Conversely, those who used Internet sites and blogs were significantly associated with the presence of such an effect. Furthermore, the responses of those who received education at junior colleges and vocational schools and above were significantly associated with the absence of such an effect. The Fukushima nuclear power plant accident was the first major nuclear disaster of the Internet age. It created an overflow of mass reporting and information. Thus, it is suggested that improving media literacy would be effective in reducing the effect of perceived DCBHR regarding nuclear accidents. A pressing task for our society is to bolster measures that will ensure the dissemination of accurate information and help users identify accurate information in extreme circumstances, such as nuclear accidents.

## Figures and Tables

**Figure 1 ijerph-20-02077-f001:**
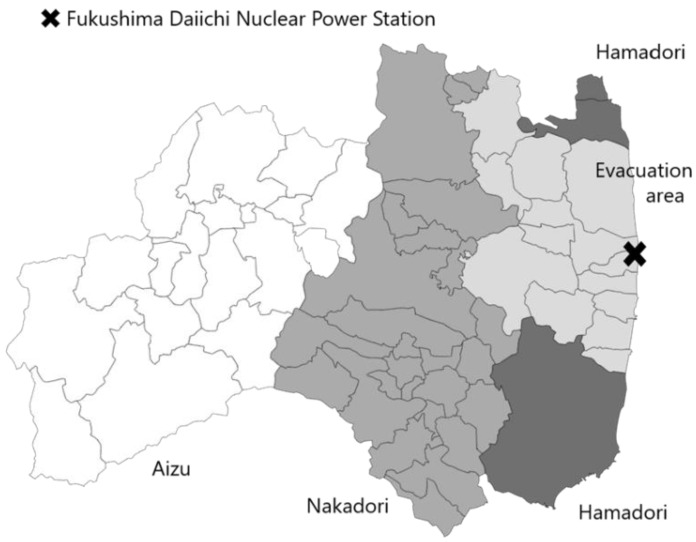
Evacuation area and areas in Fukushima. Regions colored in light gray correspond to the municipalities where evacuation orders were issued.

**Figure 2 ijerph-20-02077-f002:**
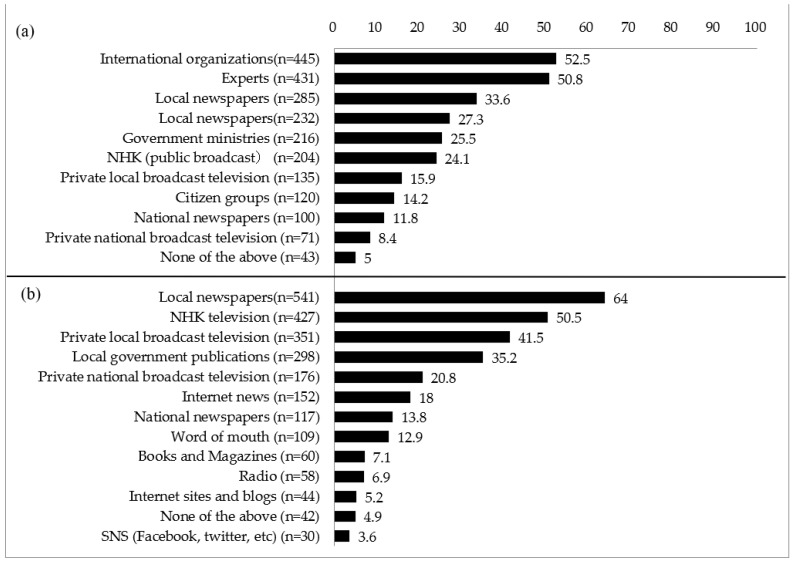
Proportion of (**a**) trusted information sources and (**b**) media used (% *n* = 849).

**Table 1 ijerph-20-02077-t001:** Basic information of the respondents.

Item	Category	*n*	%	Total
Age	Young adults (20–44 years)	201	23.7	
	Middle aged (45–64 years)	332	39.1	
	Older adults (65–80 years)	316	37.2	849
Gender	Male	380	44.8	
	Female	469	55.2	849
Area	Evacuation area	187	22.0	849
	Other (Aizu, Nakadori, Hamadori)	662	78.0	849
Education ^#^	High school or below	433	51.6	840
	Two-year college, vocational school, or above	407	48.5	840
Current family structure ^#^	Single person household	105	12.4	846
	Other	741	87.3	846
Residence ^#^	Residing at home	634	74.9	
	Not residing at home	212	25.1	846
Children at the time of the earthquake	No	387	45.6	
	Yes	462	54.4	849
Employment ^#^	Working	519	62.1	
	Not working	317	37.3	836
Neighborhood association resident association	Participating	526	62.0	849
	Not participating	323	38.0	
NPO, volunteer/citizen activity organization, co-operative association	Participating	85	10.0	849
	Not participating	764	90.0	
Vocational organizations such as business association, etc.	Participating	80	9.4	849
	Not participating	769	90.6	
Health literacy ^#^	High	424	52.4	809
	Low	385	47.6	
Food reference values ^#^	Know	341	41.1	830
	Do not know	489	58.9	

^#^ The total number does not match due to missing values. NPO: Nonprofit organization.

**Table 2 ijerph-20-02077-t002:** The distribution of responses to the question on the effect of perceived DCBHR.

	The Effect of Perceived DCBHR			
Area	Absent (*n*, %)		Present (*n*, %)	
Evacuation area	53	28.3	134	71.7
Other (Aizu, Nakadori, Hamadori)	291	44.0	371	56.0
Total	344	40.5	505	59.5
	*p* < 0.001			

**Table 3 ijerph-20-02077-t003:** Results of the univariate analysis for trusted information sources with the effect of perceived DCBHR as the response variable (*n* = 848).

Trusted Sources of Information about Radiation			The Effect of Reputational Damage				*p*
			Absent		Present		
		Total	*n*	%	*n*	%	
International organizations	no	403	169	41.9	234	58.1	0.401
	yes	445	174	39.1	271	60.9	
Experts	no	417	172	41.2	245	58.8	0.641
	yes	431	171	39.7	260	60.3	
Government ministries	no	632	240	38.0	392	62.0	0.012
	yes	216	103	47.7	113	52.3	
Local newspapers	no	563	222	39.4	341	60.6	0.397
	yes	285	121	42.5	164	57.5	
National newspapers	no	748	299	40.0	449	60.0	0.441
	yes	100	44	44.0	56	56.0	
NHK (public broadcast)	no	644	255	39.6	389	60.4	0.369
	yes	204	88	43.1	116	56.9	
Private local broadcast television	no	713	276	38.7	437	61.3	0.018
	yes	135	67	49.6	68	50.4	
Private national broadcast television	no	777	314	40.4	463	59.6	0.943
	yes	71	29	40.8	42	59.2	
Local government	no	616	255	41.4	361	58.6	0.359
	yes	232	88	37.9	144	62.1	
Citizens’ groups	no	728	297	40.8	431	59.2	0.610
	yes	120	46	38.3	74	61.7	

**Table 4 ijerph-20-02077-t004:** Results of the univariate analysis for media used by respondents with the effect of perceived DCBHR damage as the response variable (*n* = 846).

Media Used about Radiation			The Effect of Reputational Damage				*p*
			Absent		Present		
		Total	*n*	%	*n*	%	
Local newspapers	no	305	133	43.6	172	56.4	0.157
	yes	541	209	38.6	332	61.4	
National newspapers	no	729	285	39.1	444	60.9	0.049
	yes	117	57	48.7	60	51.3	
NHK television	no	419	160	38.2	259	61.8	0.189
	yes	427	182	42.6	245	57.4	
Private local broadcast television	no	495	195	39.4	300	60.6	0.468
	yes	351	147	41.9	204	58.1	
Private national broadcast television	no	670	263	39.3	407	60.7	0.175
	yes	176	79	44.9	97	55.1	
Radio	no	788	314	39.8	474	60.2	0.207
	yes	58	28	48.3	30	51.7	
Internet news	no	694	280	40.3	414	59.7	0.920
	yes	152	62	40.8	90	59.2	
Internet sites and blogs	no	802	331	41.3	471	58.7	0.032
	yes	44	11	25.0	33	75.0	
SNS (Facebook, Twitter, etc.)	no	816	326	40.0	490	60.0	0.142
	yes	30	16	53.3	14	46.7	
Books and magazines	no	786	321	40.8	465	59.2	0.374
	yes	60	21	35.0	39	65.0	
Local government publications	no	548	233	42.5	315	57.5	0.093
	yes	298	109	36.6	189	63.4	
Word of mouth	no	737	302	41.0	435	59.0	0.395
	yes	109	40	36.7	69	63.3	

**Table 5 ijerph-20-02077-t005:** Results of the univariate analysis for other items with the effect of perceived DCBHR as the response variable.

Item	Variables		The effect of the				*p*
			Absent		Present		
		Total	*n*	%	*n*	%	
Age	>44 years	648	247	38.1	401	61.9	0.011
(*n* = 849)	20–44 years	201	97	48.3	104	51.7	
Area	Non-evacuation area	662	291	44.0	371	56.0	0.000
(*n* = 849)	Evacuation area	187	53	28.3	134	71.7	
Education	Two-year college, vocational school, or above	407	181	44.5	226	55.5	0.032
(*n* = 840)	High school or below	433	161	37.2	272	62.8	
Having children at the time of the earthquake	Yes	462	164	35.5	298	64.5	0.001
(*n* = 849)	No	387	180	46.5	207	53.5	
Social participation	Neighborhood association · resident association	526	198	37.6	328	62.4	0.029
(*n* = 849)	no	323	146	45.2	177		
	Nonprofit organization, volunteer/citizen activity organization, co-operative association	85	26	30.6	59	69.4	0.049
(*n* = 849)	No	764	318	41.6	446	58.4	
	Vocational organizations such as business association, peer association, industry group, labor union, etc.	80	19	23.8	61	76.3	0.001
(*n* = 849)	No	769	325	42.3	444		
Participation	No	211	101	47.9	110	52.1	0.012
(*n* = 849)	Some	638	243	38.1	395	61.9	
Food reference values	Wrong	489	215	44.0	274	56.0	0.011
(*n* = 830)	Correct	341	120	35.2	221	64.8	

**Table 6 ijerph-20-02077-t006:** Results of the multivariable logistic regression analysis with the presence or absence of the effect of perceived DCBHR as the objective variable and trusted source of information and media consumed as the explanatory variables. (*n* = 820).

The Effect of Perceived DCBHR				
			95% Confidence Interval	
Item	OR	*p*	Lower Limit	Upper Limit
Age (20–44 years)	0.71	0.05	0.50	1.01
Gender	0.99	0.97	0.74	1.34
Trust in government ministries	0.67	0.02	0.48	0.93
Trust in a national newspaper	0.83	0.40	0.53	1.29
Trust in private local broadcast television	0.63	0.02	0.42	0.93
Used Internet sites and blogs	2.50	0.01	1.20	5.20
Evacuation area	2.17	<0.01	1.48	3.19
No children	0.62	<0.01	0.46	0.84
Junior college, vocational school, or above	0.70	0.02	0.52	0.94
Neighborhood association · resident association	1.34	0.07	0.98	1.82
NPO, volunteer/citizen activity organization, co-operative association	1.48	0.13	0.89	2.46
Vocational organizations such as business association, peer association, industry group, labor union, etc.	2.46	<0.01	1.33	4.52
Knowledge about food reference values	1.43	0.02	1.06	1.93
*R* ^2^	0.12			

NPO: Nonprofit organization.

## Data Availability

Data is not suitable for public deposition owing to ethical concerns. Researchers who have an interest in the analysis using the data, please submit requests to the Fukushima Medical University Ethics Committee (rs@fmu.ac.jp) for access to confidential data.

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
