# Peer review of "Relationship between the Effects of Perceived Damage Caused by Harmful Rumors about Fukushima after the Nuclear Accident and Information Sources and Media"

_ijerph, 2023, doi:10.3390/ijerph20032077_

Round 1
Reviewer 1 Report (Previous Reviewer 2)
The authors have significantly improved the manuscript. I have no further comments
Author Response
Thank you very much for your review.
Reviewer 2 Report (Previous Reviewer 3)
The study has improved a lot. However, there are changes to be made, especially in wording. For example, the title I would lighten it and leave it as "Relationship between the effects of rumors".
The research question What caused the rumors related to the 2011 Fukushima nuclear accident, I would change it to What was the rumor belief related to and what protected from it?
IT REMAINS UNEXPLAINED HOW THE RUMOR IS OPERATIONALIZED,. THIS POINT IS IMPORTANT.
What is measured is not the EFFECT OF THE RUMOR, BUT THE PERCEPTION OF THE INFLUENCE OF THE RUMOR. THERE ARE NO OBJECTIVE VARIABLES TO MEASURE THE EFFECT. THIS HAS TO BE CHANGED IN EVERYTHING AND IT IS A LIMITATION TO BE DISCUSSED. We measure an opinion and an attitude, not a behavior.
It is therefore important to redo the debate and the conclusions taking this into account. It would also be important to include some literature on this issue (e.g. Beléndez, M., Martín Llaguno, M., Suriá Martínez, R., & Hernández-Ruiz, A. (2004). Health worries: analysis of mass media influence; Dömötör, Z., Nordin, S., Witthöft, M., & Köteles, F. (2019). Modern health worries: A systematic review. Journal of psychosomatic research, 124, 10978).
On the other hand, I don't quite understand why they talk about anxiety if they neither measure it nor use it. Since it is not used, the paragraphs in which it is mentioned should be eliminated.
Author Response
We would like to thank Reviewer #2 for their insightful comments, which have helped us significantly improve our manuscript. We are grateful for the time and energy you have expended to help us improve our work. Please find our responses to each of your points and suggestions below. In the revised manuscript, all the changes are marked in red font.
Point1: The study has improved a lot. However, there are changes to be made, especially in wording. For example, the title I would lighten it and leave it as "Relationship between the effects of rumors".
Response 1: Thank you for your comment. We do not think changing the title to “Relationship between the effects of rumors” is possible because it would be different from the translation of Fuhyohigai as harmful rumor in the previous study. The official text issued by Fukushima Prefecture [4](Fukushima Prefecture. Vision for Revitalization in Fukushima Prefecture. 2011.) also uses the translation “damage caused by harmful rumors.”
Fuhyo, or a harmful rumor, in Japanese, is different from a rumor that usually occurs in the vicinity of a disaster-affected area. Fuhyo exists and persists long after the disaster is over, even away from the disaster area due to media coverage. These points are discussed in [2](Sekiya, 2014),(Ohtsuru et al., 2015)[5]. Describing this to be a rumor could be confused with simple interpersonal communication or word of mouth.
Therefore, we have changed the title as follows:
Relationship between the effects of damage caused by perceived harmful rumors about Fukushima after the nuclear accident and information sources and media.
Point2: The research question What caused the rumors related to the 2011 Fukushima nuclear accident, I would change it to What was the rumor belief related to and what protected from it?
Response 2: Thank you for your suggestion. However, we deemed that using the word “rumor” for the RQ would not be appropriate due to the same reason stated above in response to Point1. Therefore, we have changed the RQ as follows:
What caused the DCBHR, which Fukushima residents perceived regarding the 2011 Fukushima nuclear accident?
What resulted in the perceived DCBHR among Fukushima residents regarding the 2011 Fukushima nuclear accident?
Point3: IT REMAINS UNEXPLAINED HOW THE RUMOR IS OPERATIONALIZED,. THIS POINT IS IMPORTANT.
What is measured is not the EFFECT OF THE RUMOR, BUT THE PERCEPTION OF THE INFLUENCE OF THE RUMOR. THERE ARE NO OBJECTIVE VARIABLES TO MEASURE THE EFFECT. THIS HAS TO BE CHANGED IN EVERYTHING AND IT IS A LIMITATION TO BE DISCUSSED. We measure an opinion and an attitude, not a behavior.
It is therefore important to redo the debate and the conclusions taking this into account.
Response 3: Thank you for your comment. We have considered your point and changed all the DBCHRs that were the objective variables to perceived DBCHRs.
Point 4.
It would also be important to include some literature on this issue (e.g. Beléndez, M., Martín Llaguno, M., Suriá Martínez, R., & Hernández-Ruiz, A. (2004). Health worries: analysis of mass media influence; Dömötör, Z., Nordin, S., Witthöft, M., & Köteles, F. (2019). Modern health worries: A systematic review. Journal of psychosomatic research, 124, 10978).
Response 4.
Thank you for your valuable suggestion. We have added a sentence about the study by Beléndez et al. as follows:
Also previous study shows findings that the frequency of Internet exposure associated with the modern health worries which includes worries about radiation significantly in Spain.[27]
Point 5.
On the other hand, I don't quite understand why they talk about anxiety if they neither measure it nor use it. Since it is not used, the paragraphs in which it is mentioned should be eliminated.
Response 5,
Thank you for your valuable suggestion. We have removed the paragraph related to anxiety.
(After L.271)
This manuscript is a resubmission of an earlier submission. The following is a list of the peer review reports and author responses from that submission.
Round 1
Reviewer 1 Report
In overall, the paper is contributing to the field of risk communication. Following up the means of people living in a disaster area and how to cope with information is an important area that need further investigations. The authors call the accident an information disaster, but there is little proof of what was really the information disaster. Today's complex digital information society will allow information from all kinds of sources and one of the major challenges is to divide between facts and opinions. Therefore, it is troublesome to read that DCBHR is defined to be caused by news reports but for the study used as a frame for all kinds of information - internet sites, blogs, SNS, Facebook, Twitter may contain news reports but also very likely personal opinions. It is not clear if the authors are treating all material as news reports from mass media (also a term that needs to be defined). This leads to an overall weakness of theory within the media and communication studies to support the study and its outcomes.
The result section shows several tables and some of them, especially Table 5, shows the "raw data" for the statistics. The authors may consider if Table 5 can be shown with less data or at least be commented upon. The reader is left to do his/her own analyses.
In the Discussion section, the authors could remind the readers by spelling out OR (p.11, row 273). Also, in the discussion section, the need for media literacy is raised. It is unclear why the authors make distinctions of media literacy and online information literacy including SNS. The media and information literacy (MIL) area is a large research field and the authors could treat the area as one to make this suggestion stronger and to show that the "broader view of the types of DCBHR..." (p.13, row 361) is problematic. It is unclear why science education is emphasized in the abstract, it seems to make more sense to address the need of media and information literacy.
Reviewer 2 Report
- This is an interesting article with a lot of potential. The main problem is that it suffers from a lack of a concrete theoretical framework. The issue of rumors and dis-/misinformation is actually quite complicated, depending on sources, media, etc. Furthermore, the time this survey was conducted (2016) this issue emerged as one of the key aspects in the post truth era. As such, the authors are advised to discuss their theoretical framework in the first part of the manuscript in regard to issues related to crisis, media and rumors, as well as news and the salience of ‘fakeness’ between different types of media.
- Regarding, the empirical part of the manuscript, this is mainly based on descriptive findings of the overall study. I missed the main RQs of the study. I also missed the hypotheses for the analysis, which are usually based on the theoretical analysis and earlier studies in the field. In addition, it would be an asset to add the questionnaire tool used for the survey in an Appendix, although some description of variables is offered on pp.3-4.
- Furthermore, the conclusion needs to depict solid arguments that derive from the study. For example, the authors write: “The Fukushima nuclear power plant accident was the first nuclear disaster since the use of the internet became widespread. The internet created an overflow of mass reporting and information” (p.14, lines 412-413). However, earlier in the manuscript (p.14, lines 394-396) they write: “…because respondents tended to be relatively older, our study population included fewer users of the internet, especially SNS, which are said to be associated strongly with disinformation and the harmful rumor.” Since the latter stands – acknowledged by the authors as a limitation of this work – how does this study support such a conclusion regarding the internet?
- One minor issue. In Table 1 (p.5) it is mentioned: “Gender Male 380 44.8% 849”. I am assuming that the remaining 55.2% is Female…
Reviewer 3 Report
The manuscript deals with a very interesting subject but nevertheless suffers from important flaws.
1. State of the art
The authors explain the circumstances but do not frame the research in any academic or scientific theory. They do not pose research questions or hypotheses.
Within the framework of what they explain, I propose that they delve into two theories: disinformation and within this the concept of rumor (they say nothing and there is a very extensive bibliography) and second, the theory of the knowledge gap (or knowledge differential): how certain information impacts on individuals by virtue of experiences and prior knowledge.
Methodology
Important flaws. First, neither the confidence level nor the margin of error of the sample is explained. It cannot be published without these data
Second: they are using as a key variable a PERCEPTION, given that the question asked as they explain is the DCBHR affected your life? How does the audience distinguish between harmful rumor and true information? They need to better explain the DCBHR, the concept and the operationalization in order to get this question in.
VERY IMPORTANT:
THEY HAVE A MESS BETWEEN MEDIA AND INFORMATION SOURCES. IT IS IMPORTANT TO EXPLAIN THESE CONCEPTS, THAT THEY UNDERSTAND WHY AND HOW THEY UNDERSTAND IT. ALL INFORMATION IN A MEDIUM ALSO HAS A SOURCE. THEY DON'T SPECIFY IT OR CODIFY IT OR ASK IT. THIS QUESTION IS RELEVANT
3- Results
Don't understand the explanation they make about the exclusion of a number of questions that to me would be relevant but they don't use. If they do not use them, why do they explain them? And, why don't they use them? It is not well explained
The tables can be improved
IN CONCLUSION. THE ARTICLE REQUIRES MAJOR CHANGES IN ORDER TO BE PUBLISHED.
Round 2
Reviewer 2 Report
The authors have responded to all comments made in the first round of Review. However, three sentences in the Introduction part (p.1) do not constitute a concrete theoretical framework for an article. There are several relevant articles dealing with rumors, disinformation, etc that the authors could use to significantly enrich their work.
In addition, the RQ added on p.2 does not offer the substantial basis for the analysis that follows, nor does it justify the main theme of the manuscript which, according to the title is the "Relationship between the effects of damage caused by harmful rumors about Fukushima after the nuclear accident and information sources and media".
My suggestions is that the authors need to attend these issues prior to publication.
Reviewer 3 Report
Thank you for the response to the comments which are clearly, in my opinion, completely insufficient for the article to be published.
First of all, and despite the sentence included, the text still lacks a clear theoretical framework and a clear objective and hypothesis.
It still does not explain sufficiently and with bibliography the concept of DCBHR, which is used to establish the "objective variable" which, moreover, according to the explanations, is poorly defined since it is still a "perception" and not a fact. Whenever they speak of absence or presence of effect they should be speaking of absence or presence of perception of effect.
They still do not establish what is considered DCBHR and how it is distinguished from real information, and the reference to studies of perceptions and disinformation is not sufficient. 15 bibliographic references, for such a rich subject for the psychosocial effects of communication, is clearly insufficient.
They still do not provide the fundamental data to know the representativeness of the sample used for the survey or the formulas used to extract the information and continue to confuse the sources with the platforms, so the conclusions and the discussion are not clear.
The authors have interesting data but the article must be clearly worked on in order to be published.